# EpiLearn: A Python Library for Machine Learning in Epidemic Modeling

Zewen Liu
Department of Computer Science
Emory University
zewen.liu@emory.edu

Yunxiao Li*
Department of Computer Science
Emory University
yunxiao.li2@emory.edu

Mingyang Wei*
Department of Computer Science
Emory University
mingyang.wei@emory.edu

Guancheng Wan
Department of Computer Science
Emory University
wgc41206703@gmail.com

Max S.Y. Lau
Department of Biostatistics and
Bioinformatics
Emory University
msy.lau@emory.edu

Wei Jin
Department of Computer Science
Emory University
wei.jin@emory.edu

## ABSTRACT

*EpiLearn* is a Python toolkit developed for modeling, simulating, and analyzing epidemic data. Although there exist several packages that also deal with epidemic modeling, they are often restricted to mechanistic models or traditional statistical tools. As machine learning continues to shape the world, the gap between these packages and the latest models has become larger. To bridge the gap and inspire innovative research in epidemic modeling, *EpiLearn* not only provides support for evaluating epidemic models based on machine learning, but also incorporates comprehensive tools for analyzing epidemic data, such as simulation, visualization, transformations, etc. For the convenience of both epidemiologists and data scientists, we provide a unified framework for training and evaluation of epidemic models on two tasks: Forecasting and Source Detection. To facilitate the development of new models, *EpiLearn* follows a modular design, making it flexible and easy to use. In addition, an interactive web application is also developed to visualize the real-world or simulated epidemic data. Our package is available at https://github.com/Emory-Melody/EpiLearn.

## KEYWORDS

Epidemiology, Epidemic models, Machine learning, Neural networks, Python

**ACM Reference Format:**
Zewen Liu, Yunxiao Li, Mingyang Wei, Guancheng Wan, Max S.Y. Lau, and Wei Jin. 2024. EpiLearn: A Python Library for Machine Learning in Epidemic Modeling. In *Proceedings of the 7th epiDAMIK ACM SIGKDD International Workshop on Epidemiology meets Data Mining and Knowledge Discovery, August 26, 2024, Barcelona, Spain.* ACM, New York, NY, USA, 5 pages.

*Equal Contribution

## 1 INTRODUCTION

Data mining in epidemiology is a crucial subject in the healthcare domain, garnering increasing attention in recent years due to the COVID-19 outbreak [1, 2]. A key focus is the development of computational methods in epidemic modeling, which incorporate disease transmission mechanisms to provide insights into changing demographic health states. The diversity of data involved in epidemic modeling necessitates a broad range of tasks, including epidemic forecasting [3], simulation [4], source detection [5], intervention strategies [6], and vaccination [7].

Traditionally, knowledge-driven approaches like mechanistic models (e.g., SIR, SIS, and SEIR [8]) have been employed for epidemic modeling. These methods utilize differential equations to explicitly model relationships among population groups in different states, e.g. Suspected, Infected, and Recovered, and have demonstrated promising performance. However, over the past decade, the rapid progress in machine learning and deep learning [9, 10, 11, 12, 13, 14, 15] has paved the way for incorporating these techniques into epidemic modeling [16]. For instance, neural networks such as Cola-GNN [17] and DASTGN [18] have been developed for forecasting tasks, demonstrating superior performance compared to traditional mechanistic models. Additionally, hybrid models that integrate neural networks with mechanistic models, such as STAN [19] and EINN [20], have been proposed to leverage existing epidemiological knowledge and enhance modeling accuracy.

Given the diversity of models available, there is a growing need for a unified platform to facilitate the study and development of computational methods for epidemic modeling. While existing packages in epidemic modeling [21, 22, 23, 24, 25, 26, 27] offer tools like predefined models, simulation methods, and visualization aids, most have discontinued maintenance or failed to extend beyond traditional mechanistic models. In contrast, machine learning techniques have been thriving and demonstrating their versatility across domains [28, 29]. The trend in epidemic modeling is increasingly favoring neural and hybrid models [16, 30, 20]. Therefore, there is an urgent need to develop a machine-learning-based epidemic library that encompasses a comprehensive set of cutting-edge models and fosters further research in this field.

In an effort to bridge the chasm between machine learning and epidemic modeling, this paper introduces *EpiLearn*, a Python-based library specifically designed for data mining within the realm of

epidemiological data. The primary objective of our library is to offer a suite of user-friendly research tools that cater to both epidemiologists and data scientists alike. This endeavor is aimed at nurturing the evolution of innovative models and at enhancing the comprehension of the dynamics of epidemic spread. The core features of EpiLearn are delineated as follows:

(a) **Common Epidemiological Tasks.** The library offers robust support for two prevalent tasks in epidemic modeling: forecasting and source detection. We have integrated automated pipelines to facilitate the training and evaluation of models within these domains.

(b) **Diverse Model Architectures.** It encompasses a range of common model architectures, including Spatial, Temporal, and Spatial-Temporal baseline models, as well as exemplary epidemic models.

(c) **Data Simulation.** The library is equipped with data simulation capabilities, such as the generation of random graphs and the simulation of epidemics on these graphs.

(d) **Transformations.** A transformation module is provided to enable the processing and augmentation of epidemiological data, enhancing the flexibility and applicability of the models.

In summary, *EpiLearn* aims to streamline the research process by providing a unified framework for implementing, evaluating, and comparing various epidemic models. The library offers a modular and extensible architecture, allowing researchers to seamlessly incorporate new models, data sources, and evaluation metrics. It will facilitate collaboration and knowledge sharing among researchers, accelerate the development of new computational methods, and ultimately contribute to a better understanding and management of epidemic situations.

## 2 RELATED WORK

The advancement of epidemic modeling and analysis has been greatly facilitated by the emergence of various software packages. Notable examples include EpiModel [21], Epifit [22], EpiEstim [23], Epinet [24], EpiDynamics [25], Eir [26], and EoN [27]. EpiModel, for instance, offers a versatile framework for modeling infectious diseases, employing deterministic, stochastic, and network-based approaches to enable a thorough analysis and simulation of epidemic dynamics.

However, these established packages encounter limitations when it comes to leveraging machine learning techniques, which represents a pivotal distinction between them and *EpiLearn*. In contrast to R-based packages such as EpiModel, *EpiLearn* is constructed on Python3 and makes extensive use of PyTorch during model development and assessment. This choice provides access to the extensive resources available within the Python and PyTorch ecosystems, preserving the capacity to integrate large pre-trained models and to perform efficient fine-tuning. Additionally, unlike Eir and EoN, *EpiLearn* transcends conventional mechanistic models by offering a diverse array of spatial, temporal, and spatial-temporal models. Moreover, *EpiLearn* encompasses a broader range of epidemic tasks, including source detection, and supplies streamlined pipelines for rapid model evaluation under these tasks. Finally, we introduce an interactive web application to enhance visualization capabilities, moving beyond the provision of static figures.

## 3 OVERVIEW OF *EPILEARN*

*EpiLearn* is an open-source Python library developed primarily using Python3 and PyTorch [31], along with other common toolkits such as PyG [32], NumPy [33], and Scikit-Learn [34]. The aim of our package is to serve as a convenient tool for studying epidemic data and for building and evaluating epidemic models. In this section, we elaborate on the features provided by *EpiLearn*. The overall features are presented in Fig. 1.

### 3.1 Tasks in Epidemic Modeling

Following the taxonomy in [16], *EpiLearn* currently supports two tasks: Forecasting and Source Detection.

(a) **Forecasting.** In the forecasting task, the model uses a given length of historical data, known as the lookback window, to predict multiple future steps, referred to as the horizon. For instance, the model predicts new infections over the next three days based on statistics from the past ten days. In this setting, both temporal and spatial-temporal models can be applied.

(b) **Source Detection.** In the source detection task, the model aims to trace back the infection process and determine patient-zero in a contact graph. Some current researchers treat source detection as a classification task. The input is usually a graph at the current time, and the output is the probability of each node being the patient-zero.

In *EpiLearn*, each task is implemented as a Python class, which incorporates features such as model fitting, evaluation, and results. In Section 3.5, we further integrate each task into a pipeline.

### 3.2 Epidemic Models

*EpiLearn* supports multiple commonly used baselines as well as some epidemic-specific models. Based on the inputs, we categorize the models into spatial, temporal, and spatial-temporal models.

(a) **Temporal Models.** Temporal models process only temporal data without additional spatial information. In *EpiLearn*, we include a wide range of types, such as an auto-regression model: ARIMA [35], a machine learning model: XGBoost [36], mechanistic models: SIR, SIS, and SEIR [8], and neural networks: GRU [37], LSTM [38], and DLinear [39]. Additionally, we incorporate an epidemic-informed neural network model [20].

(b) **Spatial Models.** Spatial models utilize static features and static graphs as inputs. Common spatial baselines include GCN [40], GAT [41], and GIN [42]. Currently, we use these models primarily for source detection tasks.

(c) **Spatial-Temporal Models.** Spatial-temporal models incorporate both spatial and temporal information during inference. *EpiLearn* provides not only baselines but also several epidemic models developed in the past three years. For common baselines, we include CNNRNN [43], DCRNN [44], ST-GCN [45], and GraphWaveNet [46]. For epidemic models, we include Cola-GNN [17], STAN [19], MepoGNN [47], EpiGNN [48], EpiCo-laGNN [49], and DASTGN [18]. These models use dynamic features as input, although some can only process static graphs.

For each type of model, *EpiLearn* provides a unified framework for training and inference.

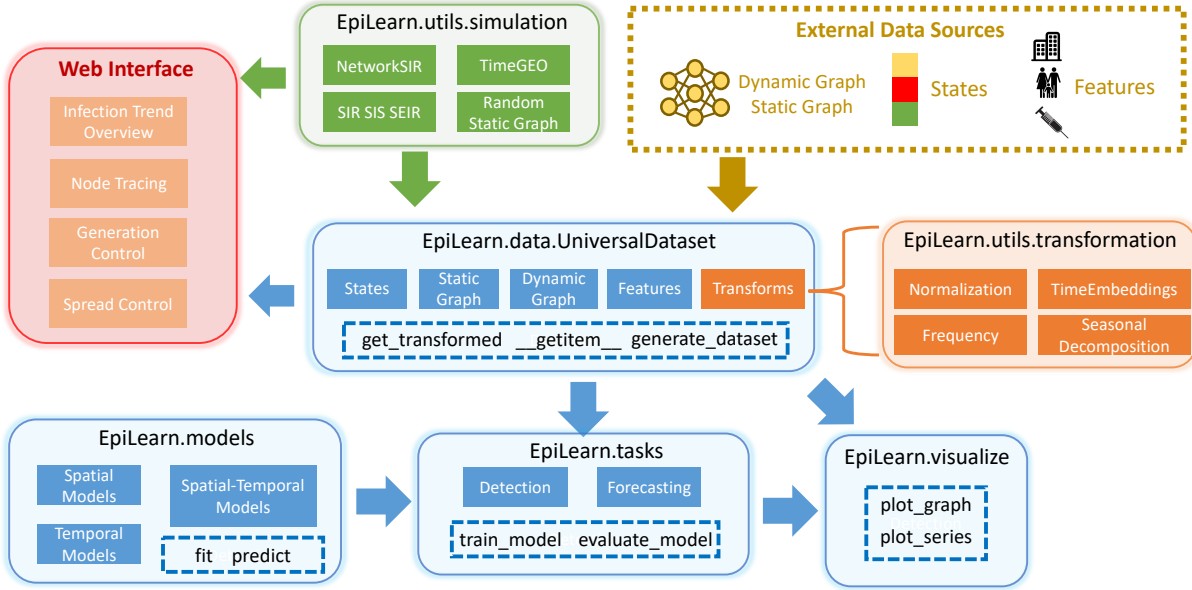

**Figure 1: Overview of *EpiLearn*.**

## 3.3 Data Processing

In addition to models, *EpiLearn* provides tools for pre-processing epidemic data. Currently, we have implemented normalization methods for both features and graph structures. Additionally, we offer useful transformations, such as converting features from the time domain to the frequency domain, adding time embeddings, and performing seasonal decomposition [50].

To ensure easy access to these data processing methods, *EpiLearn* follows a design similar to PyTorch Vision. During the pipeline construction, users can simply call the *compose* function and add the desired transformations as inputs.

## 3.4 Data Simulation

Similar to some R packages for epidemiology, *EpiLearn* provides simulation tools for quickly generating epidemic data. Specifically, we offer three types of simulations: spatial simulation, temporal simulation, and spatial-temporal simulation.

For spatial data, *EpiLearn* utilizes functions from PyG to generate random static graphs. When node features are available, node connections can be established by calculating the cosine similarity among nodes. For temporal data, *EpiLearn* uses mechanistic models introduced in Section 3.2 to simulate future health states over a given number of steps. For spatial-temporal data, *EpiLearn* employs TimeGEO [51] and NetworkSIR to simulate edge weights and node features across all regions over time.

These simulation tools enable users to create realistic epidemic scenarios for testing and validating models, facilitating research and application in epidemic modeling and analysis.

## 3.5 Epidemic Modeling Pipeline

To accelerate the training and evaluation of models on different datasets, *EpiLearn* provides a streamlined pipeline for various tasks, as illustrated in Fig. 1. At the beginning of the pipeline, we initialize the dataset with node features and states, static graphs, and dynamic graphs. Next, transformations are applied to the dataset.

For different tasks, we introduce a specific task class, such as forecasting, and input a prototype of the current model and dataset. Finally, we use the *train* function to train the current model on the dataset and the *evaluate* function to perform the evaluation.

## 3.6 Code Demonstration

To further show the convenience of *EpiLearn*, we provide a short demonstration below, which illustrates the building of the forecasting task. With a few lines of code listed below, we can easily train and evaluate a given model for forecasting tasks. This decoupling design of the task, dataset, and model also enables flexible changes made to the data or model, allowing swift evaluation on multiple models or datasets.

```python
from epilearn.models.SpatialTemporal.STGCN import STGCN
from epilearn.data import UniversalDataset
from epilearn.utils import transforms
from epilearn.tasks.forecast import Forecast
# initialize settings
lookback = 12 # inputs size
horizon = 3 # predicts size
# load toy dataset
dataset = UniversalDataset()
dataset.load_toy_dataset()
# Adding Transformations
transformation = transforms.Compose({
    "features":[transforms.normalize_feat()],
    'graph': [transforms.normalize_adj()]})
dataset.transforms = transformation
# Initialize Task
task = Forecast(STGCN, lookback, horizon)
# Training
result = task.train_model(dataset=dataset, loss='mse')
# Evaluation
evaluation = task.evaluate_model()
```

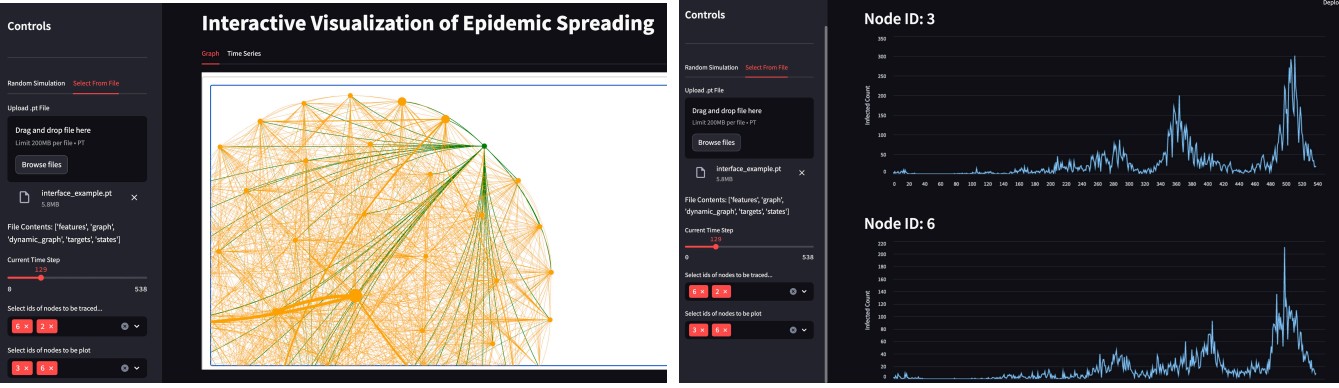

**Figure 2: The interface developed for visualizing epidemic data.**

## 3.7 Interactive Application

For further convenience, *EpiLearn* also builds a web application to visualize the epidemic data. At the current stage, we provide visualization of graph data and provide tools like tracing and showing variations of particular nodes over time. In addition, the web application also provides a simulation of the spread of the epidemic based on NetworkSIR. A demonstration is shown in Fig. 2.

## 4 KEY DESIGNS AND PROJECT FOCUS

In this section, we illustrate the key designs and focus of *EpiLearn*.

(a) **Modular and Decoupled Components.** *EpiLearn* breaks down the training and evaluation process into distinct components: model, dataset, transformations, and configurations. For every model, we provide unified training and prediction functions. The dataset component, which offers multiple features such as data splitting, requires a few key inputs, including node features and graphs. Regarding transformations, we adopt an extensible design similar to the module in PyTorch, where transformation functions are aggregated in the *Compose* function.

(b) **User-Friendly Framework.** Building on the aforementioned modules, we offer a unified framework for training and evaluating models and datasets on specific tasks. Users can easily switch among different models and datasets within our designed pipelines and modify configurations as needed. For epidemiologists and data scientists, *EpiLearn* also provides an interface to present data overviews and other useful tools like simulation.

(c) **Extensibility.** The modular design of our library also ensures extensibility. Researchers interested in testing or developing new epidemic models can follow a structure similar to the baseline models. By creating a new file for the PyTorch model or a new function for transformations, users can easily implement and test their methods.

(d) **Reproducibility.** To ensure reproducibility, we provide detailed documentation on setting up the environment, including the required dependencies and versions. We also include scripts for downloading datasets and running experiments with fixed random seeds to minimize result variations. The code and documentation for this project are publicly available at: https://github.com/Emory-Melody/EpiLearn.

## 5 IMPACTS AND FUTURE PLANS

**Impacts.** This paper presents *EpiLearn*: a Python toolkit for epidemic modeling and analysis. We provide various features, including task pipelines, data simulation, and transformations, aiming for *EpiLearn* to have a positive impact on both the machine learning and epidemiology domains.

(a) **Data Scientists' Perspective.** For data scientists, *EpiLearn* serves as a comprehensive toolbox and framework for quickly building and evaluating epidemic models. The modular design allows users to easily modify and add new features and models. Additionally, the unified pipeline facilitates fast training and testing of models on different tasks, simplifying the process of benchmarking existing models.

(b) **Epidemiologists' Perspective.** For epidemiologists, *EpiLearn* provides several epidemic models and tasks, along with simulation methods and interactive applications for visualizing epidemic data. These tools assist users in understanding the dynamics of complex systems, testing empirical hypotheses about outbreak trajectories, and gaining an overview of interventions such as vaccinations and quarantines.

**Future Plans.** Looking ahead, we plan to expand *EpiLearn* to include more general epidemic tasks such as projection and surveillance, along with related baselines and epidemic models. Continuous updates to data processing tools will enhance model training and data analysis. Furthermore, additional simulation functions will be developed to aid decision-making during epidemic crises. Lastly, ongoing development of our web application aims to provide a more comprehensive and user-friendly analysis experience.

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
