# OpenReview forum: "EpiLearn: A Python Library for Machine Learning in Epidemic Modeling"
_KDD.org/2024/Workshop/epiDAMIK — KDD 2024 Workshop epiDAMIK_

### Official Review · Reviewer_VJog · 2024-06-28
**Great initiative but missing experimental validation**

**Rating:** 2
**Confidence:** 5

**Review:**

The authors propose a new Python based library to encompass common epidemic analytics tasks.

The authors main contribution are around making available a claimed easy to use python package that can be a key tool for epidemic modeling by researchers. Specifically, the authors provide access to many state of the art machine learning based spatial/temporal/spatio-temporal models that can be trained/evaluated on user data. Such models can be directly used for a couple of pre-defined epidemic tasks and further analyzed using standard techniques such as simulation in a dynamic web server. This In contrast to many currently available packages that (a) either mainly focus on classical models such as mechanistic models or (b) don't provide a holistic toolkit for end to end analysis.

The package has been developed as a modular component, with seemingly similar design patterns to scikit-learn, that can be extended by users.

While the codebase does show that a significant effort has been put in, the manuscript lacks from experimental validation of the impact of the package. The authors seem to claim that the package is easy to use - however, beyond a toy dataset example the paper lacks experimental validation of the package being able to either (a) study a new epidemic problem and/or (b) reproduce results from previously published works. Case studies on such problems are crucial to investigate among other things the usability and performance (both computational efficiency and task performance) of the package.

Similarly, the author have identified two different set of users for their package - however no user studies have been presented to justify the claims.

Finally, while the package has been described in some details, the selected design pattern has not been articulated in a formal manner.


Overall, this is a promising work but as a scientific publication might need further experimental validation

---

### Official Review · Reviewer_1Pkk · 2024-07-01
**Great tool and impressive work!**

**Rating:** 4
**Confidence:** 3

**Review:**

Summary :

The authors of this paper develop a Python package called ``EpiLearn'', which supports two main tasks in epdiemic modeling : (1) Forecasting and (2) Source Detection.

Review :

The pros of this work heavily outweigh the cons. I believe that the current github repo of EpiLearn (including the visualization tool) is extremely intuitive for any developer who is familiar with Github and its workings. This is mainly because of the documentation provided here (https://vermillion-malasada-a2864e.netlify.app/html/index.html). The visualization is quite interpretable to novices and I love the fact that it is very user-friendly :-)
The paper is very well-written and Figure 1 (i.e. The overview of EpiLearn) clearly explains the pipeline of how EpiLearn works.

One main concern that I had was why only these two stages (Forecasting and Source Detection) were focused in this paper. I think we must holistically think of the problem of epidemic modeling (i.e. from end-to-end), starting from the data collection stage. While data collection and data preprocessing do exist in your pipeline (i.e. `EpiLearn.utils.transformation'), I think the paper should also incorporate a major focus on data collection (i.e. how one must collect such data, how do you obtain such permits to collect, what kind of data augmentation strategies could be employed for certain kind of data etc).
(But again, this is beyond the scope of a workshop paper and I will not let this hinder my score :)

This is a great, monumental work in making both a data scientist's and an epidemiologist's life easier. I hope to see this project active for the many years to come :)

---

### Official Review · Reviewer_D1LZ · 2024-07-02

**Rating:** 4
**Confidence:** 5

**Review:**

Summary:
The paper introduces a new open-source library for Epidemic modeling that includes some of the recent Deep learning and GNN based models. The library also provides example datasets and modular pipeline for data pre-processing and visualization.

Strengths:
1. The library is easy to use due to modular approach
2. It contains recent GNN based and deep learning models which support spatio-temporal forecasting

Weaknesses:
1. The library is not very extensive. It is not clear why authors chose specific models over others. Perhaps they can characterize their methods using a broader taxonomy of models [1].

Reference:
[1] Rodríguez, Alexander, et al. "Data-centric epidemic forecasting: A survey." arXiv preprint arXiv:2207.09370 (2022).

---

### Official Review · Reviewer_SRrP · 2024-07-02
**Review of the paper - EpiLearn: A Python Library for Machine Learning in Epidemic Modeling**

**Rating:** 4
**Confidence:** 4

**Review:**

Summary: In this paper, the authors design a Python toolkit "EpiLearn" for epidemic modeling by integrating ML techniques for forecasting and source detection - they specifically highlight its flexibility and easy model development and evaluation.

Quality and clarity:  the paper is quite well written and I appreciate the detailed presentation in this paper. Example code and figures given are very clear and the github repository is provided for detailed tutorial. However, some key things are not mentioned in the paper - for instance are the forecasts point or probability? What evaluation metrics are included?

Originality and significance: The Epilearn library could be valuable tool for researchers studying epidemics and wok with Python. It can be used to develop new models and understand how diseases spread. Thus the key features include - tools for forecasting and source identification, ability to incorporate epidemic simulations.  Overall, it promises to benefit both researchers and public health efforts.

Pros: 1. consider spatial and temporal models
2. ability to simulate epidemic scenarios
3. flexibiility of selecting lookback window and forecasting steps
4. repoducible - code and detailed tutorials available in git

Cons: Current version supports only two tasks - forecasting and source detection.
Potential extensions could include -  integrating additional signals to forecast and better understand epidemic dynamics.
Another direction for improvement could involve incorporating interventions with simulated data.